# Reported Cases and Diagnostics of Occupational Insect Allergy: A Systematic Review

**DOI:** 10.3390/ijms24010086

**Published:** 2022-12-21

**Authors:** Eva Ganseman, Mieke Gouwy, Dominique M. A. Bullens, Christine Breynaert, Rik Schrijvers, Paul Proost

**Affiliations:** 1Laboratory of Molecular Immunology, Research Group Immunity and Inflammation, Department of Microbiology, Immunology and Transplantation, Rega Institute, KU Leuven, 3000 Leuven, Belgium; 2Allergy and Clinical Immunology Research Group, Department of Microbiology, Immunology and Transplantation, KU Leuven, 3000 Leuven, Belgium; 3Department of Pediatrics, University Hospitals Leuven, 3000 Leuven, Belgium; 4Department General Internal Medicine–Allergy and Clinical Immunology, University Hospitals Leuven, 3000 Leuven, Belgium

**Keywords:** insects, occupational, allergy, asthma, rhinitis

## Abstract

A significant part of adult-onset asthma is caused by occupational exposure to both high- and low-molecular-mass agents. Insects are occasionally described to cause occupational allergy in professions including anglers and fishers, laboratory workers, employees of aquaculture companies, farmers, bakers, sericulture workers and pet shop workers. Occupational insect allergies are often respiratory, causing asthma or rhinoconjunctivitis, but can be cutaneous as well. The European Union recently approved three insect species for human consumption, enabling an industry to develop where more employees could be exposed to insect products. This review overviews knowledge on occupational insect allergy risks and the tools used to diagnose employees. Despite the limited availability of commercial occupational insect allergy diagnostics, 60.9% of 164 included reports used skin prick tests and 63.4% of reports used specific IgE tests. In 21.9% of reports, a more elaborate diagnosis of occupational asthma was made by specific inhalation challenges or peak expiratory flow measurements at the workplace. In some work environments, 57% of employees were sensitized, and no less than 60% of employees reported work-related symptoms. Further development and optimization of specific diagnostics, together with strong primary prevention, may be vital to the health conditions of workers in the developing insect industry.

## 1. Introduction

A European survey revealed that occupational exposure causes 10–25% of adult-onset asthma [1]. Occupational asthma can be caused by an irritant or a sensitizing agent [2]. In the latter, the sensitizing agent is taken up by antigen presenting cells in the airway, processed and presented to naïve T cells, which then differentiate into Th2 cells. The Th2 response will result in production of immunoglobulin E (IgE) by B-cells, and this IgE will be bound by the high affinity FcεRI receptor on effector cells: mast cells and basophils. Upon second exposure by the sensitizing agent, effector cells may release mediators such as histamine, leukotrienes, various chemokines and cytokines, leading to symptomatic allergic asthma [3,4]. Both high- and low-molecular-mass agents are associated with a higher incidence in asthma in employees, and the highest relative risk for new onset asthma was observed in nurses and cleaners [1].

A less known and less frequent trigger for occupational asthma and allergy are insects. Employees can be subdivided into those with direct contact with insects during insect-rearing or handling, and those with accidental insect exposure due to infestation of the workplace [5]. The following occupations have been connected to respiratory insect allergies: anglers and fishers, laboratory workers, employees of aquaculture companies, farmers, bakers, sericulture workers and pet shop workers [6,7]. Occupational insect allergies are usually provoked by the wings, legs, setae, scales or feces of the insect, and although occupational insect allergies are often respiratory, they can be cutaneous as well [8].

Today, occupational insect allergy has been described occasionally, but recently, a new interest in insects has emerged in the European Union: edible insects for human consumption or animal feed. Insects could offer a solution to an increased demand in protein due to a growing world population and a change in diets towards higher protein contents [9]. Insects have a beneficial nutrient composition, produce low levels of greenhouse gasses and are very efficient in converting feed into body mass [9,10,11]. Insects are considered novel foods in the European Union and thus require European approval to enter the market. The European Union has now authorized three insect species for human consumption: the yellow mealworm (or *Tenebrio molitor*), the migratory locust (or *Locusta migratoria*) and the house cricket (or *Acheta domesticus*) [12,13,14]. Consequently, an increased exposure to insects among employees can be expected.

To diagnose occupational asthma with certainty, a specific inhalation challenge (SIC) or serial measurements of the peak expiratory flow (PEF) with work-related changes are needed. Probable occupational asthma can be defined by evidence for asthma by a post-bronchodilator test or a non-specific bronchial hyperreactivity test, combined with a specific IgE (sIgE) or skin prick test (SPT) for the culprit insect, as described before [15,16]. Using these diagnostic criteria, a large study on post-hire asthma in insect-rearing workers revealed the incidence of post-hire asthma to be 16.2 per 1000 person-years compared to 9.2 in office workers (1000 person-years would equal following 1000 persons for 1 year). Exposure to *Lepidoptera* species further increased the incidence of post-hire asthma to 26.9 per 1000 person-years [17]. In this systematic review, we aim to give an overview of all reports on occupational insect allergy, combined with an overview of the diagnostics that were used to support the allergies.

## 2. Methods

The literature overview started with a search for “Occupational insect allergy” in Pubmed, including reports in English only (Figure 1). A total of 289 hits were screened for relevance and 139 publications were withheld. All publications could be classified in the following phylogenetic orders: *Orthoptera, Coleoptera, Lepidoptera, Blattodea, Diptera, Hemiptera, Hymenoptera* and *Psocoptera*. An additional Pubmed search was conducted for each phylogenetic order; for example, “*Orthoptera* allergy”. If the search resulted in more than 1000 publications (e.g., “*Blattodea* allergy”), the search was narrowed to occupational allergies within this phylogenetic order (e.g., “occupational *Blattodea* allergy”). For insect species that were mentioned in the selected publications, an additional search was conducted using the non-scientific name of the insect, for example: “Grasshopper allergy”. This last search was performed in the Pubmed, Scopus and Ovid databases. Screening of all manuscripts was performed by manuscript title and abstract. Studies that did not indicate clearly which patients or patient samples were included, or that did not indicate to cover occupational allergy, were excluded. The publications were only used when written in English. Abstracts that were solely published for scientific conferences were excluded if no supporting information was available. Reviews were not included. Publications that exclusively described therapeutic strategies for occupational insect allergy, without including diagnostics, were excluded. In the Ovid search results, only publications of high relevance (five stars, as defined by the database) were considered. An overview of the literature search can be found in Figure 1.

## 3. Results

### 3.1. Orthoptera

#### 3.1.1. Grasshopper Allergy

In 1953, Frankland et al. discovered that 50% of locust breeders had a positive locust SPT (Table 1) [18]. Twenty years later, the same work place was reinvestigated and results confirmed a significant risk for occupational allergy: 26% suffered from work-related wheeze, 35% from work-related rhinitis and 33% from work-related contact urticaria [19]. Also, another follow-up study at this work environment pointed to multiple locust allergens with varying molecular mass: 18, 29, 37, 43, 54, 66 and 68 kDa. Moreover, by inhibition experiments it was shown that there was limited inhibition of locust sIgE by house dust mite (HDM) extracts, reducing the potential impact of cross-sensitization on the study. The same study performed air sampling in the locust-rearing rooms for 72 h, and the allergens present could efficiently inhibit locust sIgE [20]. Another cross-sectional study of 10 laboratory workers noted that 60% of employees experienced work-related symptoms, and the major IgE binding proteins had a molecular mass of 30, 33, 35 or 70 kDa. Additionally, the study did not find a difference in allergenicity between the wings, feces or body of the locust [21], although others suggest that the peritrophic membrane might be the culprit [20].

In one patient, locust allergy was confirmed by a bronchial provocation with locust extract that caused a 23% drop in forced expiratory volume within 1 s (FEV1) ten minutes after inhalation [25]. Another case report showed *Locusta migratoria* sIgE, although the patient was only exposed to *Schistocerca gregaria*, suggesting cross-reactivity between both species [24]. Not only respiratory symptoms are of concern, as patients can suffer from severe contact dermatitis [26], sometimes well before progressing towards respiratory disease [22]. Recently, Wang et al. identified Hexamerin-like protein 2 as the main allergen in locust allergy by using pooled sera of 10 sensitized employees, the first identified locust allergen [23].

#### 3.1.2. Cricket Allergy

Although many reports focus on food allergy to crickets, and its strong relation with shrimp allergy [34], reports on occupational allergy to crickets are to date rare. The first report of occupational cricket allergy was made on two laboratory workers, working in an amphibian facility where they fed crickets to laboratory animals [31]. Both workers were diagnosed by SPT, bronchial provocation, sIgE and leukocyte histamine release tests, all performed using experimental cricket extracts, but the cricket species was not further defined. Both patients had to leave their job eventually, due to the inability to control the allergic symptoms. Another patient developed allergy upon exposure to field crickets or *Acheta campestris* in a pet store and reacted to multiple allergens of varying molecular mass: 17, 32, 47 and 62 kDa [27]. A common allergen across four species of crickets (*Acheta domesticus, Gryllodes sigillatus, Gryllus bimaculatus* and *Gryllus assimilis*) was identified as hexamerin-like protein 2 by de Las Marinas et al. [30] in two additional patients, one of them a cricket breeder. By using serum samples of laboratory personnel exposed to *Acheta domesticus*, Francis et al. showed that arginine kinase, a pan-allergen, is an allergen in crickets as well [28]. Additionally, a zoo owner developed generalized dermatitis to the feed of exotic birds, which contained crickets, and he tested positive for crickets in SPT [33]. One additional patient could be diagnosed with occupational cricket allergy with certainty by peak expiratory flow measurements that confirmed a work-related pattern combined with sIgE, although the species is unknown [32].

### 3.2. Coleoptera

#### 3.2.1. Beetle Allergy

Beetles can be pests in legumes, such as lentils and peas, where they are an occupational risk to the farmers, agronomists or even to the cooks preparing the food (Table 2) [35,36,37]. The use of extracts of non-infested legumes, infested legumes and the beetles *Bruchus lentis* or *Bruchus pisorum* in SPT and bronchial challenge tests could exclude allergy to the legumes themselves. Another common allergenic beetle is the grain weevil or *Sitophilus granaries*, which is, as the name implies, a pest in different grains, leading to two case reports already in the 1960s [38,39]. Following these results, a larger study of millworkers found a positive grain weevil SPT in 57% of them, whereas one-third noted a productive cough for at least three weeks each year [40]. An additional study identified 66 bakers with positive SPT for the grain weevil, and although the pest can only thrive in whole grain, it looks like the allergenicity persists in the processed (cleaned and milled) stored grain [41].

Some beetles, such as *Trogoderma variabile*, can infest almost anything but prefer wheat, barley or rice and are resistant to extreme temperature changes, making it a very persistent pest. In a nasal provocation test, a strong decrease in cross-sectional area and nasal volume was observed after challenging a pet food manufacturer with *Trogoderma variabele* extract. No less than 15 protein bands were recognized by the patient’s IgE [42]. *Dermestidae*, a family within the *Coleoptera*, was shown to be allergenic as early as 1941, when a museum curator was exposed to the beetle during the preparation of skeletons for display in the museum. The occupational asthma of the curator was confirmed by SPT, although the exact species of *Dermestidae* was undefined [43]. Allergenicity of *Dermestidae* was proven to a greater extent in a wool worker by SPT, sIgE and conjunctival and bronchial provocation and immunoblots, which showed a wide variety of IgE binding proteins [44].

**Table 2 ijms-24-00086-t002:** *Coleoptera* allergy.

Beetle Allergy					
Species	1st Author	Year	#Cases	Occupation	Diagnostics
*Bruchus lentis*	Armentia A [37]	2003	1	Agronomist	SPT, SIC, immunoblot
*Bruchus lentis*	Armentia A [35]	2006	16	Farmers, cooks	SPT, SIC, oral provocation, sIgE
*Bruchus pisorum*	Armentia A [36]	2020	6	Farmers, agronomists	SPT, patch test, SIC, oral provocation, sIgE, immunoblot
*Dermestidae*	Brito FF [44]	2002	1	Wool worker	SPT, SIC, conjunctival provocation, sIgE, immunoblot
*Dermestidae*	Sheldon JM [43]	1941	1	Museum curator	SPT
*Sitophilus granarius*	Herling C [41]	1995	66	Bakers	SPT, sIgE, immunoblot
*Sitophilus granarius*	Lunn JA [40]	1966	75	Millworkers	SPT, SIC
*Sitophilus granarius*	Frankland AW [39]	1964	2	Laboratory	SPT
*Sitophilus granarius*	Lunn JA [38]	1966	1	Laboratory	SPT, SIC
*Trogoderma variabile*	Bernstein JA [42]	2009	1	Pet food manufacturer	Reversibility test, SPT, nasal provocation, sIgE, immunoblot
** *Tenebrionidae* ** **Allergy**					
**Species**	**1st Author**	**Year**	**#Cases**	**Occupation**	**Diagnostics**
*Alphitobius diaperinus*	Schroeckenstein DC [45]	1988	3	Laboratory	SPT, sIgE, inhibition test, HRT
*Tenebrio molitor*	Bernstein DI [46]	1983	5	LFB	SPT, sIgE
*Tenebrio molitor*	Siracusa A [47]	1994	14	LFB	SPT, sIgE PEF measurements, BHR, inhibition test
*Tenebrio molitor*	Armentia A [48]	1997	50	Cereal workers	SPT, sIgE, conjunctival provocation, SIC
*Tenebrio molitor*	Bernstein J [49]	2002	1	Teacher	SPT, nasal provocation
*Tenebrio molitor*	Siracusa A [50]	2003	76	LFB	SPT, sIgE
*Tenebrio molitor*	Panzani R [51]	2008	54	Bakers	SPT, BHR
*Tenebrio molitor*, *Zophobas morio*	Renström A [7]	2011	59	Pet shop	sIgE, spirometry
*Tenebrio molitor*	Broekman HCHP [52]	2017	4	Breeders	SPT, sIgE, BAT, immunoblot, DBPCFC
*Tenebrio molitor*	Francis F [28]	2019	31	Laboratory	SPT, sIgE, immunoblot
*Tenebrio molitor*	Nebbia S [53]	2019	2	Food industry	SPT, sIgE, immunoblot, BAT
*Tenebrio molitor*	Ganseman E [54]	2022	1	Laboratory	SPT, sIgE, immunoblot, BAT, inhibition test
*Tenebrio molitor*, *Alphitobius diaperinus*	Schroeckenstein DC [55]	1989	1	Animal handler	SPT, sIgE, immunoblot, inhibition test
*Tribolium confusum*	Schultze-Werninghaus G [56]	1991	125	Flour	sIgE, inhibition test, immunoblot
*Tribolium confusum*	Alanko A [57]	2000	1	Flour	SPT, sIgE, SIC
*Zophobas morio*	Bregnbak D [33]	2013	1	Zoo owner	SPT

BAT: basophil activation test, BHR: bronchial hyperreactivity test, DBPCFC: double-blind placebo-controlled food challenge, HRT: histamine release test, LFB: live fish bait, PEF: peak expiratory flow, SIC: specific inhalation challenge, sIgE: specific IgE, SPT: skin prick test, #cases indicates the number of employees assessed for occupational allergy.

#### 3.2.2. Tenebrionidae Allergy

*Tenebrionidae*, or black beetles, is a family that consists of an estimated 2000 species. One of them, *Tribolium confusum*, or the confused flour beetle, has proven to be allergenic in workers exposed to flour (Table 2). One non-atopic 35-year-old male developed IgE-mediated allergy after 4 years of exposure to old flour, in which the pest could thrive [57]. In a larger study, nine bakers showed sIgE to *Tribolium confusum* after exposure to rye and wheat flour, and this sIgE could not be inhibited by HDM, rye or wheat flour [56].

Another source of beetle exposure can be found in the production and usage of live fish bait. Siracusa et al. noted work-related symptoms in 9.2% and sensitization in 31.6% of 76 employees exposed to live fish bait, including *Tenebrio molitor*, or the yellow mealworm, as one of the species of interest [50]. A previous study by Siracusa et al. studied 14 subjects who reported work-related respiratory symptoms. They found evidence of mealworm sensitization by a radioallergosorbent test (RAST, a sIgE test) and SPT in three of those [47]. Moreover, in a warehouse handling mealworms as live fish bait, four out of five workers reported immediate-onset asthma, rhinitis or contact urticaria, and two asthmatic workers were diagnosed by bronchial provocation tests [46].

Not only live fish bait poses risk to employees; the yellow and lesser mealworm (*Alphitobius diaperinus*) are bred for scientific purposes as well, where they proved to be an allergenic risk [45,55]. Cross-reactivity has been observed between both species, as an employee allergic to the lesser mealworm showed sensitization to yellow mealworms as well in sIgE tests and sIgE inhibition [45]. On the other hand, cross-reactivity was not observed between the yellow mealworm and the house cricket, as shown with serum samples of exposed laboratory personnel [28]. Another example of mealworm allergy was found in a teacher keeping mealworms in the classroom to educate children about the life cycle of the insect. The teacher was diagnosed by a positive mealworm SPT and nasal provocation test [49]. Additionally, the super worm or *Zophobas morio* elicited severe dermatitis in a zoo owner, as shown by SPT [33].

### 3.3. Lepidoptera

#### 3.3.1. Moth Allergy

One baker developed occupational allergy to another pest in flour: the flour moth or *Ephestia keuhniella* (Table 3). The baker’s sIgE to the flour moth could be strongly inhibited by HDM, evidencing a cross-reactivity between mites and the flour moth [58]. A second study observed a positive SPT for *Ephestia* (species not further defined) in 15 farmers and bakers, although this did not correlate with *Ephestia* sIgE [59]. The flour moth can be beneficial too, as the eggs are used to feed predatory mites which in turn are used in biological control in greenhouses or crops. A laboratory employee breeding the eggs developed occupational asthma, also co-occurring with HDM sensitization [60].

In the live fish bait industry, the greater wax moth can be of concern as in one study, where 57.1% of employees who experienced work-related symptoms had a positive SPT for the greater wax moth, although sIgE was positive in only 42.9% [50]. An additional employee with a greater wax moth allergy showed cross-sensitization to other *Lepidoptera* species such as owlet moths and silk moths [61]. One additional patient, a zoo owner who fed exotic birds with the greater wax moth, developed severe dermatitis and had a positive SPT [33]. Sometimes, a heavy infestation with moths, for example the Douglas fir tussock moth, can cause a transient occupational risk as many wood-logging employees noted itching skin and eyes, runny nose and sometimes asthma during two consecutive summers. Important to note is that the study was mainly questionnaire-based [62].

The silk moth, or *Bombyx mori*, is by far the most reported as an occupational risk. The larva of the silk moth creates a cocoon that consists of a single 600 to 900 m strand that is used to produce silk. A first study noted at least one respiratory symptom (cough, sputum, shortness of breath, wheeze or tightness of chest) in 56.6% of workers, with a potential for occupational asthma in 33.9% [63]. In an additional large cross-sectional study, 36% of 243 workers reported occupational asthma. In the same population, 21.8% of workers had a positive SPT to both cocoon and pupal allergens [64]. Another study observed a positive SPT for the cocoon of the moth in 35% of silk workers, but in the control group 17.5% also elicited a positive cocoon SPT, raising questions about the specificity of the SPTs used in these studies [65]. Using the serum of 24 subjects with occupational asthma to the silk worm, 13 allergens were identified by Zou et al., with the main focus on Bom m 9, a 30 kDa protein precursor [66].

**Table 3 ijms-24-00086-t003:** *Lepidoptera* allergy.

Moth Allergy					
Species	1st Author	Year	#Cases	Occupation	Diagnostics
*Bombyx mori*	Harindranath N [64]	1985	243	Silk industry	SPT, sIgE
*Bombyx mori*	Uragoda CG [63]	1991	53	Silk industry	PEF measurements, questionnaire
*Bombyx mori*	Gowda G [65]	2014	120	Silk industry	Reversibility test, SPT
*Bombyx mori*	Zuo J [66]	2015	24	Unknown	Immunoblot, inhibition test
*Ephestia*	Armentia A [59]	2004	15	Baker, farmer	SPT, SIC, sIgE, inhibition test
*Ephestia*	Panzani R [51]	2008	57	Bakers	SPT, BHR
*Ephestia kuehniella*	Mäkinen-Kiljunen S [58]	2003	1	Baker	SPT, sIgE, inhibition test, nasal provocation
*Ephestia kuehniella*	Moreno Escobosa MC [60]	2014	1	Laboratory	SPT, sIgE, immunoblot
*Galleria mellonella*	Stevenson DD [61]	1966	1	LFB	SPT, HRT, immunoblot, SIC
*Galleria mellonella*	Siracusa A [47]	1994	14	LFB	SPT, sIgE, PEF measurements, BHR, inhibition test
*Galleria mellonella*	Siracusa A [50]	2003	76	LFB	SPT, sIgE
*Galleria mellonella*	Bregnbak D [33]	2013	1	Zoo owner	SPT
*Lymantria dispar*, *Pectinophora gossypiella*, *Euproctis chrysorrhoea*	Suarthana E [17]	2012	157	Insect breeders	sIgE
*Orgyia pseudotsugata*	Press E [62]	1977	428	Timber, forestry workers	SPT, questionnaire
**Caterpillar Allergy**					
**Species**	**1st Author**	**Year**	**#Cases**	**Occupation**	**Diagnostics**
*Lymantria dispar*, *Orgyia pseudotsugata*	Etkind P [67]	1982	17	Laboratory	Scratch test
*Thaumetopoe pityocampa*	Vega JM [68]	1997	1	Pine-forest worker	SPT, immunoblot
*Thaumetopoe pityocampa*	Vega JM [69]	1999	55	Pine-forest workers	SPT, immunoblot
*Thaumetopoe pityocampa*	Vega JM [70]	2000	16	Pine-forest workers	SPT, immunoblot
*Thaumetopoe pityocampa*	Rebollo S [71]	2002	13	Unknown	SPT, immunoblot
*Thaumetopoe pityocampa*	Vega J [72]	2004	30	Pine-forest workers	SPT, immunoblot
*Thaumetopoe pityocampa*	Morales-Cabeza C [73]	2016	1	Pine-resin worker	SPT, sIgE, BAT, immunoblot
*Thaumetopoe pityocampa*	Ricciardi L [74]	2021	3	Pine-forest workers	Questionnaire

BAT: basophil activation test, BHR: bronchial hyperreactivity, HRT: histamine release test, PEF: peak expiratory flow, SIC: specific inhalation challenge, sIgE: specific IgE, SPT: skin prick test, #cases indicates the number of employees assessed for occupational allergy.

#### 3.3.2. Caterpillar Allergy

Caterpillars are known to cause a variety of skin manifestations, including contact urticaria via non-immunological or hitherto unknown mechanisms, sometimes called Lepidopterism [75]. Nonetheless, Vega et al. showed that allergic mechanisms can also play a role, as 87.5% of the included pine-forest workers had a positive SPT and detectable sIgE in immunoblots towards the processionary caterpillar (*Thaumetopoe pityocampa*) (Table 3). Five potential allergens were pinpointed, four of them of low molecular mass (<18 kDa) [69,70], further confirmed by an additional study [71]. Moreover, different exposure groups were investigated and pine-cone and resin collectors, farmers and stock breeders were found to be at higher risk compared to forestry personnel, construction workers, residential gardeners or entomologists [72]. Caterpillar allergy can be severe, as a pine-forest worker experienced a sudden onset rash that extended to the whole body, combined with tong oedema, weakness, shortness of breath and nausea after disturbing a pine processionary caterpillar nest. The patient had a positive SPT, while 36 controls remained negative, and showed IgE binding to two proteins of 25 and 35 kDa [68]. Another patient, with underlying mastocytosis, experienced an anaphylactic reaction after contact with the pine caterpillar at his job as a pine-resin worker. The patient had a positive SPT for the caterpillar and 86.5% of basophils degranulated upon stimulation with caterpillar extract [73]. Beside pine-forest workers, an additional study of laboratory personnel, exposed to the Gypsy moth caterpillar, tested them for sensitization using SPT: 88.2% of workers had a positive test for the Gypsy moth caterpillar. The authors do acknowledge that more research is needed to prove these reactions are caused by IgE-mediated mechanisms [67].

### 3.4. Blattodea

#### Cockroach Allergy

Environmental exposure to cockroaches is known to cause sensitization and asthma by infesting living areas and homes. The most studied species are the American cockroach, or *Periplaneta americana*, and the German cockroach, or *Blatella germanica* [76]. It is no surprise that the same species are known to cause occupational allergy (Table 4) [77], for example in laboratory personnel in whom the nasal patency dropped by 69.2% after a nasal provocation with extracts of the American cockroach [78]. Additionally, the German cockroach was shown to infest seagoing ships, causing an occupational risk to seamen, as it was shown that 29.6% of seamen were sensitized according to a cockroach SPT and 52.8% according to cockroach sIgE [79,80]. Cockroaches do not only infest seagoing ships or living areas, but stored cereal too, which again endangers bakers and stored cereal workers [48,81]. Although cockroach allergy is often respiratory, the saliva of *Blaberus giganteus*, or the giant cockroach, has been reported to induce immediate pruritis, redness and whealing of the skin. Nonetheless, sIgE towards the whole body of the cockroach was higher than the sIgE towards the saliva. The patient did develop rhinitis and asthma eventually [82]. Moreover, a zoo owner had a positive cockroach SPT, species undefined, after developing dermatitis after feeding exotic birds [33].

### 3.5. Diptera 

#### 3.5.1. Fly Allergy

The fruit fly, or *Drosophila melanogaster*, is a widely used laboratory insect, often in genetics research (Table 5). A large study on 286 workers in contact with the fruit fly divided them into four categories: high and frequent exposure, high and infrequent exposure, low and frequent exposure and low and infrequent exposure. In all 286 workers, the sensitization rate to *Drosophila melanogaster* was 6%, but in the high and frequent exposure group, 15.4% of workers were sensitized [83]. In another study, 7 of 22 employees (31.9%) reported work-related symptoms, and for 6 of them, evidence for fruit fly allergy was found by SPT, sIgE or bronchial provocation tests [84]. In six fruit fly-allergic employees, diagnosed by the presence of respiratory symptoms combined with a positive SPT for the *Drosophila* adult or larva stage and/or positive sIgE against *Drosophila*, evidence pointed to hexamerin to be a potential allergen [85]. One additional allergic worker, diagnosed by SPT and nasal provocation, showed IgE binding to hexamerin too, combined with tropomyosin, alcohol dehydrogenase and sarcoplasmic calcium-binding protein 1 of the fruit fly [86]. The Mediterranean fruit fly was also allergenic as extensively shown by SPT, sIgE, bronchial provocation, peak expiratory flow measurements during the work day and immunoblotting in two employees involved in production of the insect [87].

Despite the prevalence of the common house fly or *Musca domestica*, occupational allergy remains rare: three separate cases were reported on allergic rhinoconjunctivitis in farmers and laboratory personnel [88,89,90], with one of them confirmed by a conjunctival provocation test [90]. By inhibition experiments, one of the patients showed no cross-reactivity to other fly species (blowfly, fruit fly and lesser house fly) [89]. One of those, the blowfly, or *Lucilia cuprina*, is a pest, and for that reason studied in a laboratory context. Three different studies by the same investigators showed an allergic potential in all developmental stages of the insect. A total of 28% of the exposed workers reported allergic symptoms, mainly of the upper respiratory system and eyes, whereas two-thirds of them actually had blowfly sIgE. Moreover, cross-reactivity with screwflies, and potentially other insects, was observed, indicating potential insect pan-allergy [91,92,93].

In an employee of mushroom cultivation, asthma and rhinoconjunctivitis turned out to be caused by the flies infesting the cultivation areas, as shown by conjunctival provocation test, sIgE, SPT and immunoblotting. The flies were named ‘champignon flies’ and further defined to be part of the *Phoridae* and *Sciaridae* families, although the exact species are unknown [94]. Another person experienced rhinoconjunctivitis after working in a strongly elk fly-infested forest during his job as a geological researcher. Both nasal and conjunctival provocation confirmed the occupational allergy [95]. In a sewage plant, a worker developed occupational asthma to the sewer fly, as confirmed by bronchial provocation, SPT and in vitro leukocyte histamine release [96]. The screwworm fly was an occupational hazard too, for example in eradication personnel [97,98]. Live fish bait handlers are known to be at risk for allergy to a variety of species including *Lucilia caesar* and *Calliphora vomitoria*. In two studies, sensitization to both species was more common among symptomatic live fish bait handlers, compared to the other insects the handlers were exposed to [47,50].

Respiratory occupational fly allergy is not the only concern, as one live fish bait handler showed contact dermatitis confirmed by SPT to *Calliphora vomitoria* larva during packaging [99]. One case of an anaphylactic reaction was reported, after a bite of the tsetse fly, or *Glossina morsitans*, in a PhD student that was shown to have IgE to the insect by dot blots [100].

**Table 5 ijms-24-00086-t005:** *Diptera* allergy.

Fly Allergy					
Species	1st Author	Year	#Cases	Occupation	Diagnostics
*Calliphora vomitoria*	Pazzaglia M [99]	2003	1	LFB	SPT
*Calliphora vomitoria*	Siracusa A [50]	2003	75	LFB	SPT, sIgE
*Ceratitis capitata*	de Las Marinas MD [87]	2014	2	Production	SPT, sIgE, SIC, immunoblot, FeNO, BHR
Champignon flies	Cimarra M [94]	1999	1	Mushroom cultivator	SPT, conjunctival provocation, PEF measurements, immunoblot
*Drosophila melanogaster*	Colomb S [85]	2017	59	Laboratory	Questionnaire, SPT, sIgE, immunoblot
*Drosophila melanogaster*	Jones M [83]	2017	286	Laboratory	Questionnaire, sIgE
*Drosophila melanogaster*	Betancor D [86]	2021	1	Laboratory	SPT, FeNO, nasal provocation, immunoblot
*Drosophila melanogaster*	Spieksma FT [84]	1986	22	Laboratory	SPT, sIgE, inhibition test, SIC
Elk fly	Laukkanen A [95]	2005	1	Geological researcher	SPT, sIgE, inhibition test, nasal and conjunctival provocation
*Glossina morsitans*	Stevens WJ [100]	1996	1	Laboratory	sIgE
*Lucilia caesar*	Siracusa A [47]	1994	14	LFB	SPT, sIgE, PEF measurements, BHR, inhibition test
*Lucilia cuprina*	Kaufman GL [92]	1986	1	Laboratory	SPT, sIgE, BHR
*Lucilia cuprina*	Baldo BA [91]	1989	30	Laboratory	sIgE, immunoblot
*Lucilia cuprina*	Kaufman GL [93]	1989	53	Laboratory	Questionnaire, sIgE
*Musca domestica*	Tee RD [88]	1985	1	Laboratory	SPT, sIgE, inhibition test
*Musca domestica*	Wahl R [90]	1997	1	Farmer	Conjunctival provocation, sIgE, immunoblot
*Musca domestica*	Focke M [89]	2003	1	Farmer	SPT, sIgE, immunoblot, inhibition test
Screwworm fly	Herrmann GH [98]	1966		Unknown	Unknown
Screwworm fly	Dille JR [97]	1968		Unknown	Unknown
Sewer flies	Gold BL [96]	1985	1	Sewage worker	SPT, HRT, SIC
**Midge Allergy**					
**Species**	**1st Author**	**Year**	**#Cases**	**Occupation**	**Diagnostics**
Chironomid midges	Baur X [101]	1992	85	Fish food, laboratory	SPT, sIgE
Chironomid midges	Teranishi H [102]	1995	1	Environmental researcher	sIgE, inhibition test, immunoblot
Chironomid midges	Seldén AI [103]	2013	8	Sewage workers	FeNO, sIgE
*Chironomus lewisi,* *Chironomus riparius*	Tee RD [104]	1985	26	Unknown	SPT, sIgE, inhibition test
*Chironomus thummi*	Liebers V [105]	1993	225	Fish food	SPT, sIgE
*Chironomus thummi*	Galindo PA [106]	1999	4	Fish food	SPT, sIgE, conjunctival and nasal provocation, immunoblot
*Chironomus thummi*	Meseguer Arce J [107]	2013	8	Fish food	SPT, PEF measurements, BHR, nasal provocation, sIgE, immunoblot, inhibition test
*Chironomus: thummi, annularius, tentans* and *tepperi*	Baur X [108]	1982	99	Fish food	SPT, sIgE, inhibition test, SIC

BHR: bronchial hyperreactivity test, FeNO: fractional exhaled, HRT: histamine release test, LFB: live fish bait, nitric oxide, PEF: peak expiratory flow, SIC: specific inhalation challenge, sIgE: specific IgE, SPT: skin prick test, #cases indicates the number of employees assessed for occupational allergy.

#### 3.5.2. Midge Allergy

Fish food often contains lyophilized larva of chironomid midges, which is a family of more than 6000 species (Table 5). Workers exposed to the fish food often develop respiratory, conjunctival or cutaneous symptoms, for which the hemoglobin of the insect is thought to be the main culprit [108]. Different species of midges seem to be cross-reactive, as Sudanese patients were sensitized to *Chironomus thummi* as tested by SPT, despite only having natural exposure to *Chironomus lewisi* [104]. Some studies found 24.7% of occupationally exposed subjects to be sensitized to the hemoglobin of chironomid midges, or Chi t 1 [101]. Besides evident occupational allergy, it was shown that many patients are sensitized to chironomids even without apparent exposure, pointing to cross-sensitization presumably by house dust mite or crustacean allergy. Patients without exposure did not show strong IgE binding towards the main allergen hemoglobin, whereas patients with true occupational allergy did [106].

In contrast, others showed that in employees working with fish food, IgE binding to the red midge could not be inhibited by *Dermatophagoides pteronyssinus* and tropomyosin was not involved. In general, the time before onset of work-related symptoms was short; for some employees, only 3 months [107]. On top of that, midges find a great breeding spot in indoor sewage water pools in Nordic countries, creating a vast exposure in sewage workers, and a first pilot study showed midge sensitization via sIgE in 38% of them. Furthermore, an environmental researcher developed rhinoconjunctivitis to adult chironomids during collection of the insect around lakes. Although adult chironomids are expected to lose most of their hemoglobins, Teranishi et al. showed that hemoglobins are potentially the responsible allergens in adult chironomid allergy too [102].

### 3.6. Hemiptera

#### 3.6.1. True Bug Allergy

True bugs are sometimes used as pest control in greenhouses, as is the case for the predatory bugs *Macrolophus pygmaeus* and *Macrolophus caliginosus* (Table 6). By sIgE measurements, 46% of greenhouse workers were shown to be sensitized to *Macrolophus pygmaeus* [109]. A more comprehensive case investigation was performed by the same research group, showing occupational allergy to *Macrolophus caliginosus* by SPT, sIgE and bronchial provocation tests [110]. True bugs can be pests themselves, as the work environment of a water bottler was infested by ground bugs, and the employee developed rhinoconjunctivitis and asthma. Occupational allergy to the ground bug was confirmed by a conjunctival provocation test, SPT, sIgE and immunoblotting [111]. As for many phylogenetic families within the *Insecta* class, true bugs are also known to infest stored grains. Fifteen workers exposed to stored grain were sensitized and seven of them had a positive bronchial provocation for *Eurygaster*, a genus within the *Hemiptera* insects [59].

#### 3.6.2. Lice Allergy

A rather special case of occupational insect allergy is allergy towards the cochineal lice, or *Dactylopius coccus* (Table 6). More specifically, it is an allergy towards carmine, the red dye produced by the lice. The dye consists of the chemical compound carminic acid and residual lice material, and is often used in food and cosmetics, where it is known to cause food allergy and contact dermatitis. One report of occupational allergy was made when a soldier had an anaphylactic reaction when a make-up stick containing carmine was used in a casualty simulation, although there was no diagnostic follow-up to confirm the allergy [112]. A few years earlier, Burge et al. had shown that carmine could cause respiratory allergy in two employees, one in cosmetics blending and one in a dye factory, by bronchial provocation tests [113].

Occupational carmine allergy has extensively been studied in dye factories by SPT, bronchial provocation tests, sIgE tests, immunoblots and inhibition tests [114,115,116]. A study of 24 workers of a dye factory found carmine sensitization in 46% of them and occupational asthma in 8.3%. As much as 29% of employees had bronchial hyperreactivity, although this can be caused by non-occupational asthma [114]. Three allergens were described in employees of dye factories: 17, 28 and 50 kDa proteins [115]. Quirce et al. found that the protein fraction of carmine with a molecular mass above 10 kDa and below 30 kDa could inhibit carmine sIgE best [116]. In a screen-printing worker, carmine (10 µg/mL) could degranulate 29% of the worker’s basophils, confirming IgE-mediated allergy [117]. Another population at risk for occupational carmine allergy are employees exposed to spices, which are often given their red color by carmine, such as butchers or workers of spice warehouses [118,119,120]. Despite first reports indicating low-molecular-mass proteins to be responsible, others reported proteins of high molecular mass (40 to over 97 kDa) [118,120], raising questions about the responsible allergens. In carmine-induced food allergy, one 38 kDa allergen has been identified as CC38K [121], but more evidence is emerging that a hapten-carrier effect of carminic acid to proteins could be detrimental [122,123,124].

**Table 6 ijms-24-00086-t006:** *Hemiptera* allergy.

True Bug Allergy					
Species	1st Author	Year	#Cases	Occupation	Diagnostics
*Eurygaster*	Armentia A [59]	2004	15	Stored grain	SPT, SIC, sIgE, inhibition test
*Eurygaster*	Panzani R [51]	2008	57	Bakers	SPT, BHR
*Macrolophus caliginosus*	Lindström I [110]	2017	2	Greenhouse workers	SPT, sIgE, SIC, reversibility test
*Macrolophus pygmaeus*	Suojalehto H [109]	2021	117	Greenhouse workers	sIgE, FeNO
*MetopopIax ditomoides*, *Microplax albofasciato*	García Lázaro MA [111]	1997	1	Water bottling	SPT, conjunctival provocation, BHR, PEF measurements, sIgE, immunoblot
Whitefly	Campion KM [125]	2012	26	Insect breeders	sIgE
**Lice Allergy**					
**Species**	**1st Author**	**Year**	**#Cases**	**Occupation**	**Diagnostics**
*Dactylopius coccus*	Burge PS [113]	1979	2	Dye factory, Cosmetics blender	SIC
*Dactylopius coccus*	Park GR [112]	1981	1	Soldier	None
*Dactylopius coccus*	Quirce S [116]	1993	9	Dye factory	SPT, sIgE, inhibition tests, SIC, oral provocation
*Dactylopius coccus*	Acero S [120]	1998	1	Spice warehouse	SPT, SIC, immunoblot
*Dactylopius coccus*	Lizaso MT [115]	2000	3	Dye factory	SPT, SIC, immunoblot
*Dactylopius coccus*	Añíbarro B [119]	2003	2	Butchers	SPT, SIC, immunoblot
*Dactylopius coccus*	Tabar-Purroy AI [114]	2003	2	Dye factory	SPT, SIC, immunoblot
*Dactylopius coccus*	Ferrer A [118]	2005	1	Butcher	SPT, SIC, immunoblot, sIgE, inhibition test
*Dactylopius coccus*	Cox CE [117]	2012	1	Screen printer	SPT, sIgE, BAT

BHR: bronchial hyperreactivity test, FeNO: fractional exhaled nitric oxide, PEF: peak expiratory flow, SIC: specific inhalation challenge, sIgE: specific IgE, SPT: skin prick test, #cases indicates the number of employees assessed for occupational allergy.

### 3.7. Hymenoptera 

#### 3.7.1. General Hymenoptera Venom Allergy

*Hymenoptera* venom allergy has a prevalence of approximately 5% in the general population [126,127]. Of all patients with *Hymenoptera* venom allergy, 17% had at least one allergic reaction at work, mostly reported in beekeepers, gardeners, firemen or forest rangers (Table 7) [126]. Sting incidence was reported to be 98.1% in Japanese forest workers [128] and 59% in pest control operators [129]. Italian forestry workers reported a sting incidence of 59% to 87%, depending on the geographical location within the country. A total of 13% of them reported large local reactions and 9% even systemic reactions after a *Hymenoptera* sting [130]. In an Israeli population, 44% reported adverse reactions of their *Hymenoptera* venom allergy on their occupational activities [131], confirmed by an Italian study of 181 patients, where 17% reported work disability [132]. 

#### 3.7.2. Honey Bee Venom Allergy

Honey bee venom (HBV) allergy is a known hazard in beekeepers and can range from mild local reactions to severe anaphylactic reactions (Table 7). Beekeeping can be both occupational or a hobby, but as reports often do not specify the extent of the beekeeping, we did not distinguish between both. When asked for their history of sting exposure, 89.9% of 494 beekeepers reported being stung in the last 12 months and 5.7% reported emergency admissions in the past [133]. In large questionnaire-based studies (460 and 1053 participants), 2.9% to 4.4% reported systemic reactions upon honey bee stings [134,135], while others reported systemic reactions to be far more prevalent, in 20% to 26% of beekeepers (200 and 218 participants), respectively [136,137]. A prospective study followed 35 new beekeepers over a 5 year time period and found that 28.6% became sensitized, most of them within the first 18 months of beekeeping [138].

In 200 beekeepers, 42% had positive sIgE levels to HBV, regardless of the number of stings [137]. When looking at the skin reactions of sensitized beekeepers, it was shown that not all beekeepers had high histamine levels in response to a sting, and histamine levels reversely correlated with LTC_4_ and LTE_4_ levels. This could be one of many explanations for varying clinical reactions after stings [139]. One study found a positive correlation between the number of stings and the sIgE towards HBV and a component of HBV, phospholipase A (PLA), and higher levels were connected to the susceptibility to systemic reactions [140]. PLA is the most abundant protein within HBV, leading to generally high levels of sIgE towards PLA [141]. After a sting, HBV-specific IgEs were found to rise for at least two weeks, but levels declined again at 6 months after the sting [142]. In general, the diagnosis of HBV allergy is standardized adequately; commercial HBV extracts are available for SPT and sIgE tests are available, both for the venom as well as for components of the venom. SIgE results have to be interpreted with regard to carbohydrate cross-reactive determinants that are known to be present in *Hymenoptera* venom [143].

In beekeepers, the number of stings before clinical reactions is higher than in the general population. Subsequently, when looking thoroughly at the immune response in beekeepers, HBV specific IgG was shown to be higher [144]. Another study confirmed higher levels of IgG_4_ in beekeepers tolerant to bee stings [145] or even found a positive correlation of the IgG_4_ with the number of bee stings [146]. The most efficient treatment of HBV allergy is immunotherapy that consists of consistent administration of HBV over 3–5 years, leading to loss of the hypersensitivity. IgG_4_ is known to rise upon immunotherapy, together with induced T cell tolerance [147]. One study reported on high efficiency levels of immunotherapy in beekeepers, although results are strongly biased by the fact that unsuccessful immunotherapy might lead to the termination of beekeeping as a hobby or even occupationally [148].

Next to the sting risk in beekeepers, contact dermatitis has been self-reported in 5.5% of beekeepers upon contact with honey bee products [133]. Three beekeepers experienced contact dermatitis during collection of honey or cleaning of the beehives. Patch tests were positive for propolis, a resin-like material made by honey bees to build their hives, in all three cases. One of them had an additional positive patch test to honey and beeswax [149]. One beekeeper reported rhinoconjunctivitis upon working in the hives, and the patient’s IgE bound a 13 kDa allergen present in honey bee bodies, larvae and the *Varroa* mite, a mite present in bee hives [150]. Two more employees in honey bee production showed symptoms of lower respiratory allergy such as coughing, wheezing and shortness of breath. Both had a positive SPT and sIgE to a whole-body extract of the honey bee. Although they also had a mild sensitization to HBV, RAST-inhibition experiments showed that the allergenic components of the body and the venom of the honey bee are distinct [151]. Another side product of the honey bees is royal jelly, a nutritious compound made by worker bees to support the queen bee. One employee showed a 56% drop in peak expiratory flow and a 44% drop in FEV1, all within 1 h of entering the workspace where royal jelly was processed [152]. Two additional laboratory workers experienced rhinoconjunctivitis and asthmatic symptoms while working with royal jelly powder, and the allergy was confirmed by specific inhalation challenge in one of them [153].

#### 3.7.3. Bumblebee Venom Allergy

Bumblebees are efficient pollinators and are for that purpose farmed, so they can be used in crop pollination. In greenhouse workers, 38% of employees showed to have bumblebee venom (BBV) sIgE, but systemic reactions were less common (5%) (Table 7) [154]. In bumblebee farms, the prevalence of systemic reactions to bumblebee stings was found to be 10%. A high level of cross-reactivity was observed, as subjects with a previous reaction or positive SPT for HBV were at higher risk for BBV allergy [155]. Others reported that cross-reactivity is rather observed in non-occupational cases, whereas occupational BBV allergy is more often bumblebee-specific. Additionally, even within the bumblebee family, allergens can vary significantly, as shown for *Bombus terrestris* and *Bombus pennsylvanicus* [156].

Kochuyt et al. treated patients with severe BBV allergy with HBV immunotherapy, as a standardized BBV was not yet available [155]. In other reports, this approach was unsuccessful, as patients reacted severely to the HBV [157], or the HBV immunotherapy was inefficient and patients reacted strongly when re-exposed to bumblebee stings [158]. This led to the idea that there must be allergenic components unique to the BBV [159] that makes component-resolved diagnostics highly valuable in *Hymenoptera* immunotherapy. For example, a biologist that lacked cross-reactivity with HBV, as shown by inhibition tests, was treated with an ultra-rush protocol with BBV, reaching a maintenance dose of 80 µg after five weeks of immunotherapy. Two months later, this patient was stung again at the workplace, but only a mild local reaction occurred [160]. Another study diagnosed six patients with BBV allergy by SPT and sIgE, and treated three of them with immunotherapy. The efficacy of the therapy was shown by an in-hospital sting challenge [161]. In a larger study on BBV immunotherapy in 11 patients, 6.2% of large local reactions were seen in the induction phase of the immunotherapy, whereas this dropped to 2.8% in the maintenance phase. In the induction phase, two severe reactions occurred, but overall it was concluded that BBV immunotherapy was relatively safe [162].

**Table 7 ijms-24-00086-t007:** *Hymenoptera* venom allergy.

General *Hymenoptera* Venom Allergy					
Species	1st Author	Year	#Cases	Occupation	Diagnostics
*Encarsia*	Campion KM [125]	2012	26	Insect breeders	sIgE
*Hymenoptera*	Kahan E [131]	1997	500	General	Questionnaire
*Hymenoptera*	Ono T [129]	1998	118	Pest-control operators	Questionnaire, sIgE
*Hymenoptera*	Incorvaia C [163]	2004	112	Forest workers	Questionnaire
*Hymenoptera*	Turbyville JC [164]	2013	3	Soldiers	Retrospective analysis
*Hymenoptera*	Paolocci G [132]	2014	181	General	Questionnaire
*Hymenoptera*	Voss JD [165]	2016	23	Soldiers	Retrospective analysis
*Hymenoptera*	Toletone A [126]	2017	104	Outdoor workers	Questionnaire
*Hymenoptera*	Ricciardi L [130]	2018	341	Forestry workers	Questionnaire
**Honey bee Venom Allergy**					
**Species**	**1st Author**	**Year**	**#Cases**	**Occupation**	**Diagnostics**
Honey bee	Light WC [140]	1975	34	Beekeepers	sIgE
Honey bee	Müller U [142]	1977	57	Beekeepers	sIgE
Honey bee	Bousquet J [166]	1982	250	Beekeepers	Questionnaire, sIgE
Honey bee	Kemeny DM [167]	1983	11	Beekeepers	sIgE
Honey bee	Nordvall SL [141]	1983	37	Beekeepers	sIgE
Honey bee	Reisman RE [151]	1983	2	Honey production	SPT, sIgE, inhibition
Honey bee	Bousquet J [137]	1984	176	Beekeepers	Questionnaire, SPT, sIgE
Honey bee	Lomnitzer R [168]	1986	15	Beekeepers	sIgE
Honey bee	Khan RH [169]	1991	14	Beekeepers	sIgE
Honey bee	Annila IT [170]	1997	78	Beekeepers	sIgE
Honey bee	García-Robaina JC [171]	1997	242	Beekeepers	SPT, sIgE
Honey bee	Kalyoncu AF [172]	1997	786	Beekeepers	Questionnaire, sIgE
Honey bee	Yee CJ [146]	1997	78	Beekeepers	sIgE
Honey bee	Eich-Wanger C [144]	1998	62	Beekeepers	SPT, sIgE
Honey bee	Manso EC [173]	1998	59	Beekeepers	sIgE
Honey bee	Annila IT [139]	2000	6	Beekeepers	SPT, HRT
Honey bee	Garrido-Fernandez SG [149]	2004	3	Beekeepers	Patch test, sIgE
Honey bee	Rudeschko O [150]	2004	1	Beekeeper	SPT, sIgE, immunoblot, inhibition test
Honey bee	Celikel S [133]	2006	494	Beekeepers	Questionnaire
Honey bee	Kalogeromitros D [138]	2006	35	Beekeepers	SPT, sIgE
Honey bee	Meiler F [174]	2008	10	Beekeepers	sIgE, cytokine production, T cell response
Honey bee	Münstedt K [134]	2008	1053	Beekeepers	Questionnaire
Honey bee	Münstedt K [148]	2010	73	Beekeepers	Questionnaire
Honey bee	Richter AG [175]	2011	852	Beekeepers	Questionnaire
Honey bee	Varga EM [145]	2013	10	Beekeepers	sIgE
Honey bee	von Moos S [176]	2013	96	Outdoor workers, beekeepers	Questionnaire
Honey bee	Celiksoy MH [177]	2014	301	Beekeepers	sIgE, cytokine production, T cell response
Honey bee	Gómez Torrijos E [153]	2016	2	Pharmacy laboratory	SPT, sIgE, BHR, SIC
Honey bee	Guan K [143]	2016	54	Beekeepers	SPT, sIgE
Honey bee	Li LS [152]	2016	1	Royal jelly factory	SPT, sIgE, PEF measurements, inhibition test, immunoblot
Honey bee	Matysiak J [178]	2016	30	Beekeepers	sIgE
Honey bee	Boonpiyathad T [179]	2017	15	Beekeepers	B cell characterization
Honey bee	Carballo I [180]	2017	158	Beekeepers	Questionnaire, sIgE
Honey bee	Ediger D [181]	2018	242	Beekeepers	Questionnaire
Honey bee	Demirkale ZH [135]	2020	69	Beekeepers	Questionnaire
Honey bee, wasp	Annila IT [136]	1996	191	Beekeepers	Questionnaire
**Bumblebee Venom Allergy**					
**Species**	**1st Author**	**Year**	**#Cases**	**Occupation**	**Diagnostics**
*Bombus terrestris,* *Bombus pennsylvanicus*	Hoffman DR [156]	2001	6	Bumblebee farm	sIgE, SPT, inhibition test
Bumblebee	Josef P [157]	1993	1	Bumblebee farm	SPT, sIgE
Bumblebee	Kochuyt A [155]	1993	5	Bumblebee farm	SPT, sIgE
Bumblebee	de Groot H [161]	1995	6	Bumblebee farm, greenhouse worker	SPT, sIgE, sting challenge, inhibition test
Bumblebee	Stapel SO [159]	1998	6	Bumblebee farm	sIgE, inhibition test, immunoblot
Bumblebee	de Jong NW [162]	1999	11	Bumblebee farm, greenhouse worker	SPT, sIgE
Bumblebee	Stern A [158]	2000	2	Biologists	SPT, sIgE, sting challenge
Bumblebee	Roll A [160]	2005	1	Biologist	SPT, sIgE, inhibition test
Bumblebee	Lindström I [154]	2022	121	Greenhouse workers	sIgE
**Wasp Venom Allergy**					
**Species**	**1st Author**	**Year**	**#Cases**	**Occupation**	**Diagnostics**
Hornet and paper wasp	Hayashi Y [182]	2014	1353	Forest workers, electrical facility field workers	Questionnaire, sIgE
Wasp	Pérez-Pimiento A [183]	2007	98	Unknown	Retrospective analysis
Yellow jacket wasp	Shimizu T [128]	1995	323	Forestry workers	Questionnaire, sIgE

BHR: bronchial hyperreactivity test, PEF: peak expiratory flow, SIC: specific inhalation challenge, sIgE: specific IgE, SPT: skin prick test, HRT: histamine release test, #cases indicates the number of employees assessed for occupational allergy.

#### 3.7.4. Wasp Venom Allergy 

Of all anaphylactic reactions to wasp venom, 18% are reported to be linked to occupation (Table 7). The number one occupation suffering from these reactions are gardeners [183]. In Japanese forest workers, 98.1% experience *Hymenoptera* stings and 21.8% experience hypersensitivity because of it. Of the forest workers, 40% had sIgE to hornet or wasp venom, whereas in comparison 30% of electrical plant workers and 15% of office workers had sIgE to either venoms [182]. A high prevalence of hornet and wasp venom sIgE, 6.3 and 22.3%, respectively, was further confirmed in 323 forest workers [128].

### 3.8. Psocoptera and Others

*Psocoptera*-infested books caused occupational rhinoconjunctivitis in an employee of a bookshop, and although the species was not further defined, the sensitization was confirmed by SPT (Table 8) [184]. A 33-year-old carpenter experienced asthmatic symptoms linked to the harvest of barley grains in April to June. Small insects, including *Liposcelis decolor*, or the booklouse, infested his office coming from the granary next to it. The patient was diagnosed using SPT and immunoblotting, pinpointing a 30 kDa allergen [185].

In hydroelectric plants, caddis flies are attracted for several reasons: a permanent habitat, a high-water flow and thus nutrition and the lights of the plant itself. A commercial caddis fly extract used in SPT was positive in 91% of employees who experienced work-related symptoms [186]. Another study on caddis fly allergy showed that peripheral blood mononuclear cells (PBMCs) of allergic workers produced more Th2 cytokines, although differences were not significant [187]. One additional case report established final confirmation of caddis fly allergenicity by a bronchial provocation test [188].

**Table 8 ijms-24-00086-t008:** *Psocoptera* allergy and others.

Psocoptera Allergy					
Species	1st Author	Year	#Cases	Occupation	Diagnostics
*Psocoptera*	Veraldi S [184]	2019	1	Book shop	SPT
**Others**					
**Species**	**1st Author**	**Year**	**#Cases**	**Occupation**	**Diagnostics**
*Liposcelis decolor*	Marco G [185]	2016	1	Carpenter, exposed to barley	SPT, immunoblot, BHR
Caddis fly	Warrington RJ [187]	2003	105	Hydroelectric power plant	SPT
Caddis fly	Miedinger D [188]	2010	1	Hydroelectric power plant	SIC
Caddis fly	Kraut A [186]	1994	28	Hydroelectric power plant	Questionnaire, SPT, sIgE, PEF measurements

BHR: bronchial hyperreactivity test, PEF: peak expiratory flow, SIC: specific inhalation challenge, sIgE: specific IgE, SPT: skin prick test, #cases indicates the number of employees assessed for occupational allergy.

## 4. Discussion

A total of 164 publications were included in this review, concerning eight different insect families (Figure 2), and the first report was made in 1941. For all insect families, reports have been gradually increasing; only for Hymenoptera venom allergies has there been a strong increase in reports starting from the late 1990s. Of all reports, 70.7% were made in European countries, 13.4% in North-American countries, 6.7% in Asian countries and 4.9% in Middle Eastern countries. Local climate might influence the work environments and levels of exposure. Since only few reports were made in non-European countries and many of the included studies are not cross-sectional studies, we did not make a comparison of prevalence in workers in different continents.

Commercial sIgE testing (ImmunoCAP, Thermofisher (2022 catalog) and Immulite, Siemens (2016 catalog)) is available for the following insect species that were included in this review: *Tenebrio molitor*, *Chironomus thummi*, *Periplaneta americana*, *Blatella germanica*, *Ephestia kuehniella*, *Bombyx mori*, *Tribolium confusum*, *Bombus terrestris*, *Vespula* spp. and *Apis mellifera*. Component-resolved diagnostics are available for *Apis mellifera* (Api m 1, Api m 2, Api m 3, Api m 5, Api m 10) and *Vespula* spp. (Ves v 1 and Ves v 5). Commercial extracts are available for skin prick tests (Stallergenes Greer (2020 catalog) or ALK (2022 catalog)) for *Bombyx mori*, *Periplaneta americana*, *Blattella germanica* and *Musca domestica*. 

A total of 100 out of 164 publications, or 60.9%, used an SPT to test for occupational allergy, and considering the limited availability of commercial extracts, were mostly done with in-house generated and non-standardized extracts. Of all reports 104, or 63.4%, used sIgE testing, including both the radioallergosorbent-test (RAST) used in the past and the current fluorimetric enzyme-linked immunoassay (FEIA) systems. Likewise, most sIgE tests are in-house generated and non-standardized tests. Occupational asthma was diagnosed with certainty in 36 of 164 publications (21.9%) by either specific inhalation challenge or peak expiratory flow measurements. Another five publications had a probable diagnosis of occupational insect asthma, with bronchial hyperreactivity or reversibility test combined with SPT or sIgE. In 42 publications (25.6%), at least one form of provocation testing was used, including specific inhalation challenges, and nasal and conjunctival provocation. In 15 publications (9.1%), no diagnostic tests were performed and the research was solely questionnaire-based.

By far the most reported occupation at risk for insect allergy are laboratory workers, followed by workers in the production of live fish bait or fish food (Figure 3). Insect breeders and general farmers are often reported to develop occupational insect allergies, as are workers exposed to flour or stored grains (e.g., bakers). With the recent approval of insects for human consumption and as animal feed, we have an industry that might grow strongly within the European Union in the coming decades.

This review showed that the prevalence of insect sensitization in employees can be as high as 57% for certain insect species [40]. In the same trend, work-related symptoms were reported in no less than 60% of employees exposed to certain insect species [21]. An important form of bias that could potentially influence the results of the studies included in this systematic review is the healthy worker effect. A healthy worker effect is a bias that arises from observational studies in occupational settings without an accurate control group [189]. First of all, the working population might be healthier than the general population [190], causing an underestimation of the risks for employees, i.e., the healthy worker hire effect. Additionally, employees who encounter work-related symptoms early on might have left their employment and thus would not show up in these observational occupational studies, i.e., the healthy worker survivor effect [189,191]. If the healthy worker effect has a strong effect on the studies currently available on occupational insect allergy, the true risk for employees might even be underestimated based on the reports included in this review.

Diagnosis of occupational allergy was supported by SPT and sIgE most often, almost always by in-house generated extracts or sIgE tests. These in-house generated methods are highly valuable but make comparisons between different studies difficult. Moreover, some studies showed a discrepancy between the SPT and sIgE tests. In that light, standardization and commercial availability of insect extracts for SPT and sIgE tests could benefit diagnosis of occupational insect allergy greatly. Component-resolved diagnostics for most insect species described in this review do not exist either, and more research is needed before reaching that stage. Component-resolved diagnostics could shine light on the severity of allergic reactions and cross-reactivity, as it does for many other allergenic sources (e.g., peanut allergy, pollen-food syndrome).

Accurate diagnosis is undeniably important in occupational insect allergy, but just as important is accurate prevention. Evidence does show that the risk of occupational allergy increases with exposure [192]. Lowering of exposure by accurate ventilation of workspaces and the use of personal protective equipment (facemasks, gloves, protective clothing, etc.) could lower the burden of occupational allergies on insect industries. Moreover, some of the studies within this review show a higher risk for occupational insect allergy in atopic workers, raising the question of whether screening for atopy before starting a job with high insect exposure could be beneficial.

## 5. Conclusions

In conclusion, this review offers an extensive overview of the current literature on occupational insect allergy. It revealed that insect exposure can sensitize large parts of the workforce and can cause work-related symptoms in equally large portions of employees.

## Figures and Tables

**Figure 1 ijms-24-00086-f001:**
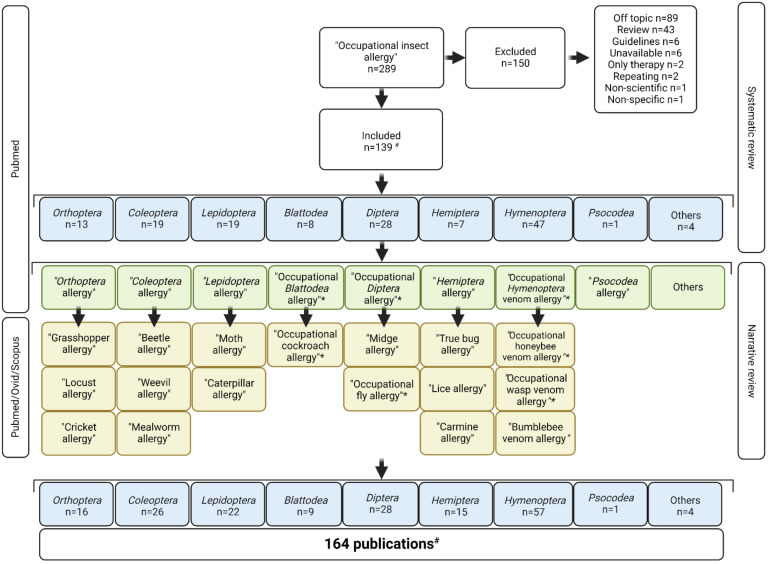
Overview of literature search. * if a search term obtained >1000 results, the search was narrowed to occupational allergies only. # some publications describe insects of multiple phylogenetic families.

**Figure 2 ijms-24-00086-f002:**
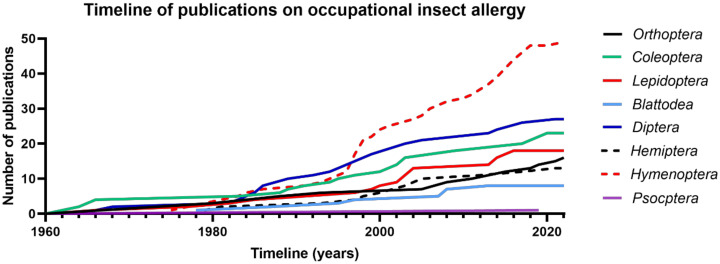
Evolution of the number of publications on occupational insect allergy. Only two publications were made before 1960 and were excluded from this graph.

**Figure 3 ijms-24-00086-f003:**
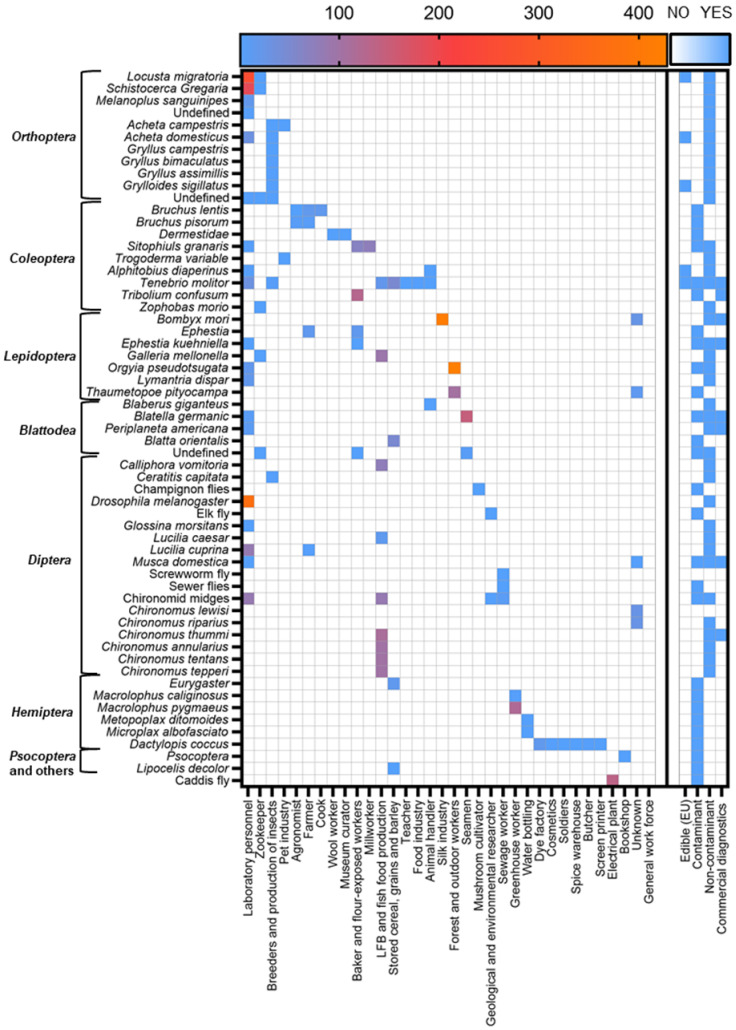
Number of cases assessed for all insect species and affected occupations. Number of cases based on the assessed cases of each publication. Edible (EU): based on current approval or applications being assessed. Exposure by insects as the contaminant vs. non-contaminant. Commercial diagnostics: either sIgE or SPT extract commercially available. Hymenoptera were excluded based on the large number of reports and currently available reviews.

**Table 1 ijms-24-00086-t001:** *Orthoptera* allergy.

Grasshopper Allergy					
Species	1st Author	Year	#Cases	Occupation	Diagnostics
*Locusta migratoria*	Lopata AL [21]	2005	10	Laboratory	SPT, sIgE, inhibition test, immunoblot
*Locusta migratoria*	Rauschenberg R [22]	2015	1	Zookeeper	SPT, sIgE
*Locusta migratoria*	Wang Y [23]	2022	57	Laboratory	SPT, sIgE, immunoblot, inhibition test
*Locusta migratoria,* *Schistocerca gregaria*	Frankland AW [18]	1953	34	Laboratory	SPT
*Locusta migratoria,* *Schistocerca gregaria*	Burge PS [19]	1980	119	Laboratory	SPT, sIgE
*Locusta migratoria,* *Schistocerca gregaria*	Tee RD [20]	1988	35	Laboratory	SPT, immunoblot, sIgE, inhibition test
*Locusta migratoria,* *Schistocerca gregaria*	Hrgovic I [24]	2018	1	Zookeeper	SPT, sIgE
*Melanoplus sanguinipes*	Soparkar GR [25]	1993	17	Laboratory	SPT, SIC
Undefined	Monk BE [26]	1988	3	Laboratory	Unknown
**Cricket Allergy**					
**Species**	**1st Author**	**Year**	**#Cases**	**Occupation**	**Diagnostics**
*Acheta campestris*	Bartra J [27]	2008	1	Pet store	SPT, nasal provocation, immunoblot
*Acheta domesticus*	Francis F [28]	2019	31	Laboratory	SPT, sIgE, immunoblot
*Acheta domesticus,* *Gryllus campestris,* *Gryllus bimaculatus*	Linares T [29]	2008	1	Cricket breeder	SPT, sIgE, immunoblot, SIC
*Gryllus assimilis,* *Gryllus bimaculatus,* *Gryllodes sigillatus,* *Acheta domesticus*	de Las Marinas MD [30]	2021	2	Cricket breeders	SPT, sIgE, immunoblot
Undefined	Bagenstose AH [31]	1980	2	Laboratory	SPT, SIC, sIgE, HRT
Undefined	Harris-Roberts J [32]	2011	32	Cricket breeders	PEF measurements, sIgE
Undefined	Bregnbak D [33]	2013	1	Zoo owner	SPT

BHR: bronchial hyperreactivity test, PEF: peak expiratory flow, SIC: specific inhalation challenge, sIgE: specific IgE, SPT: skin prick test, HRT: histamine release test. #cases indicates the number of employees assessed for occupational allergy.

**Table 4 ijms-24-00086-t004:** *Blattodea* allergy.

Cockroach Allergy					
Species	1st Author	Year	#Cases	Occupation	Diagnostics
*Blaberus giganteus*	Kanerva L [82]	1995	1	Animal care	SPT, sIgE
*Blatella germanica*	Oldenburg M [80]	2008	145	Seamen	Questionnaire, SPT, sIgE, spirometry
*Blatella germanica*, *Periplaneta americana*	Steinberg DR [78]	1987	6	Laboratory	SPT, sIgE, nasal provocation, inhibition test
*Blatta orientalis*	Armentia A [48]	1997	50	Cereal workers	SPT, sIgE, SIC, conjunctival provocation
*Blatta orientalis*	Panzani R [51]	2008	54	Bakers	SPT, BHR
*Periplaneta americana*	Zschunke E [77]	1978	4	Laboratory	Open patch test
Undefined	Marraccini P [81]	2007	1	Baker	SPT, sIgE, BHR, SIC
Undefined	Oldenburg M [79]	2008	6	Seamen	Questionnaire
Undefined	Bregnbak D [33]	2013	1	Zoo owner	SPT

BHR: bronchial hyperreactivity test, SIC: specific inhalation challenge, sIgE: specific IgE, SPT: skin prick test, #cases indicates the number of employees assessed for occupational allergy.

## Data Availability

Not applicable.

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
