# Peer review of "Reported Cases and Diagnostics of Occupational Insect Allergy: A Systematic Review"

_ijms, 2022, doi:10.3390/ijms24010086_

Round 1
Reviewer 1 Report
In my opinion, this review, by Ganseman and colleagues, is well-organized and thorough. The review has focused on occupational insect allergy with confirmed diagnosis. The descriptions of each included study are brief and clear. I have only a few questions:
1. As the authors reviewed all 164 publications, were there any differences in prevalence between different regions? For example, Europe vs. America, Africa, and Asia. Were there any differences in prevalence between different climates?
2. As I used the website Pubmed to search for “Occupational insect allergy”, I had 346 results. I understand that a few new pieces of research might be published after the authors’ work was finished, but according to Pubmed, there were only 8 articles published in 2022. The number did not match the author’s search, 289 (before excluding 139 references).
I do not mean to ask the authors to fill up the remaining; I just want to know that, did the authors used any limitations (for example, language) or MeSH terms when searching PubMed?
Minor suggestions
1. The resolution of figure 3 was low. Is there a higher resolution version?
Author Response
Reply to Comment 1:
We have checked the geographical origin of all manuscripts. 70.7% of all reports came from European countries, 13.4% came from North-America, 6.7% came from Asian countries and 4.9% from Middle eastern countries. Potentially, countries with colder climates breed or process insects in closed indoor environment with limited ventilation. We would except a similar level of indirect insect-exposure (for instance from grains, flour, cereal infestation) in all climates. Other reasons might be underreporting of occupational insect allergy in non-European countries, strengthened by only including literature in English language. We have added the geographical origin of the included reports in the discussion (L509-L515).
Since only few reports were made in non-European countries and many of the included studies are not cross-sectional studies, we did not make a comparison of prevalence in workers in different continents.
Reply to comment 2:
The only limitation used was language: only manuscripts in English were considered. As of today, this would result in 292 Pubmed results. We have added clarification in the methods (L68-69).
Reply to minor comment:
We have increased figure 3 from 300 to 600 dpi.
Reviewer 2 Report
Excellent work.
The Authors developed a very smart and elegant approach to the literature, based on the systematic screening and additional search supplementing it. Huge effort of several top specialists with a fantastic output.
I have no reservations. In the very case of this work, any possible change would make it only worse.
Author Response
We thank the reviewer for the positive comments.
Reviewer 3 Report
This is a nicely written review on occupational insect allergy and the authors have done a great job of summarizing results from various relevant literature.
Following are a few comments:
1. Authors should provide a brief paragraph at the beginning of the article about the basic immunological mechanism of allergic response in context with the article.
2. Please recheck all the figures some of the numbers are not adding up e.g. in figure 1 after exclusion there are 139 studies indicated however if you add the numbers on the following it comes out as 146.
Author Response
Comment1: Authors should provide a brief paragraph at the beginning of the article about the basic immunological mechanism of allergic response in context with the article.
Answer: We have added a small paragraph on basic allergology L33-L39.
Comment 2: Please recheck all the figures some of the numbers are not adding up e.g. in figure 1 after exclusion there are 139 studies indicated however if you add the numbers on the following it comes out as 146.
Answer: This apparent inconsistency results from the fact that one publication can describe occupational allergy to insects of multiple phylogenetic families and thus such a publication could be added to the total of multiple phylogenetic families. This was indicated by the # in the legend of Fig.1. Within the 164 publications, four publications described 2 insect families, two publications described 3 insect families and two publications described 4 insect families. This explains the differences.